# Can Naloxegol Therapy Improve Quality of Life in Patients with Advanced Cancer?

**DOI:** 10.3390/cancers13225736

**Published:** 2021-11-16

**Authors:** Rita Ostan, Giuseppe Gambino, Italo Malavasi, Gianluca Ronga, Maria Solipaca, Michela Spunghi, Silvia Varani, Raffaella Pannuti, Enrico Ruggeri

**Affiliations:** National Tumor Assistance (ANT), Via Jacopo di Paolo, 36-40128 Bologna, Italy; rita.ostan@ant.it (R.O.); giuseppe.gambino@ant.it (G.G.); italo.malavasi@ant.it (I.M.); gianluca.ronga@ant.it (G.R.); maria.solipaca@ant.it (M.S.); michela.spunghi@ant.it (M.S.); silvia.varani@ant.it (S.V.); raffaella.pannuti@ant.it (R.P.)

**Keywords:** cancer, naloxegol, opioid-induced constipation, palliative care, antagonists, opioid receptor, quality of life

## Abstract

**Simple Summary:**

The frequent use of opioid in-home palliative care settings is associated with a high incidence of opioid-induced constipation (OIC) that can significantly worsen a cancer patient’s quality of life. Laxative therapy is the first-line in OIC treatment, and although its application is recommended from the beginning of opioid therapy, in most cases it does not resolve the symptoms. Naloxegol is the specific target therapy for opioid-induced constipation. To our knowledge this is the first study aimed at evaluating the impact of a naloxegol therapy on the quality of life of advanced cancer patients with a short or very short life expectancy in a home palliative setting.

**Abstract:**

This observational study aims to evaluate the efficacy of naloxegol therapy in resolving opioid-induced constipation (OIC) and in improving the quality of life in a home palliative care cancer setting. Advanced cancer patients with OIC (Rome IV criteria) not relieved by laxatives started a naloxegol therapy 25 mg/day for 4 weeks. Quality of life was evaluated by Patient Assessment of Constipation Quality-of-Life (PAC-QoL) at day 0 and day 28; background pain by Numerical Rating Scale, number of weekly spontaneous bowel movements and Bowel Function Index (BFI) were evaluated at day 0 and every week. Seventy-eight patients who completed the 4-week study improved all four PAC-QoL dimensions (physical and psychological discomfort, worries/concerns and satisfaction level). Weekly spontaneous bowel movements increased and BFI improved. Background pain reduced after seven days and remained lower during the following weeks. Seventy-two patients dropped out the study before day 28 with a reduced survival compared to patients completing the study. Even in these patients, an improvement of bowel function was observed after two weeks. Naloxegol was effective in improving the quality of life, resolving OIC and reducing overall pain in patients with advanced cancer.

## 1. Introduction

Opioids represent the most frequently used and effective pain therapy in cancer care settings [1]. The specific activation of μ-opioids receptors modifies gastrointestinal functions by reducing peristalsis, increasing sphincter tone, increasing fluid reabsorption and decreasing its secretion, with the consequent onset of constipation [2,3,4,5], called opioid-induced constipation (OIC).

OIC is present in up to 90% of cancer patients, and occurs from the beginning of analgesic treatment [6,7]. OIC causes discomfort in the patient and negatively affects the quality of life [8]. The pain from constipation is added to that from the cancer itself, inducing anxiety and urgency in the patient to resolve the constipation.

OIC is generally treated with hygiene/dietary measures and the use of laxatives [9], which in most cases do not resolve the symptoms [10,11,12]. Therefore, the doctor or the patients themselves may be led to reduce the opioid therapy, without resolving constipation but only producing a worsening of cancer pain [13].

The definition of OIC and its identification have benefited greatly from the implementation of the Rome IV criteria [14,15]. Constipation is often underestimated by the doctor and the patient [16,17], and therefore requires careful evaluation of both the objective and subjective symptoms. Opioids can induce OIC or exacerbate a pre-existing functional constipation and this should motivate the choice of the specific therapeutic line [18]. Laxative therapy is the first-line in OIC treatment [19,20], and its application is recommended from the beginning of opioid therapy (Figure 1). If laxative therapy is not effective, a change to target therapy with peripheral acting μ-opioid receptor antagonists (PAMORAs) is required [19,20,21,22]. Naloxegol, a PEGylated naloxone derivative, was approved by the European Medicines Agency (EMA) in 2014 for the treatment of OIC in patients with or without cancer. A recent study showed the efficacy of naloxegol on symptoms and quality of life in cancer patients with OIC in real life settings [23,24]. Compared with other PAMORAs, naloxegol showed a higher compliance by the patients (easy intake by assumption of one tablet/die), and a reduced passage across the blood-brain barrier by its PEGylation [20].

The objective of this study was to evaluate if naloxegol therapy can improve the quality of life by reducing OIC in a home palliative-care setting for advanced cancer patients.

## 2. Materials and Methods

### 2.1. Setting

The home care model managed by the ANT Foundation employs a hospital-at-home approach in which a multidisciplinary team of physicians, nurses and psychologists, all trained in palliative care, work around-the-clock, 24 h/7 days a week to assist cancer patients. The service is free for the patients and it is offered in agreement with the National Health Service.

### 2.2. Study Design and Patients

This observational prospective study was conducted between September 2018 and December 2020 in full accordance with the Declaration of Helsinki and the Good Clinical Practice. Participants were recruited in 9 Italian cities among patients assisted at home by the ANT Foundation.

The study enrolled patients with advanced cancer (unlikely to be cured or controlled with treatment) and OIC, defined according to Rome IV criteria [25], not relieved by first-line laxatives. The palliative care physician verified the presence of OIC and recommended to the family doctor the prescription of a therapy with naloxegol, according to clinical practice [26]. The patients were enrolled in the study if they met the following inclusion criteria: (1) advanced cancer patients aged ≥18 years maintaining their mental capacity (able to understand the aim and procedures of the study, to sign the informed consent and to answer the questionnaires), (2) clinical indication for starting a therapy with 25 mg/day naloxegol according to the Italian Medicines Agency (AIFA) [26], i.e., opioid therapy for at least 2 weeks and inadequate response to laxatives for at least 3 days. The exclusion criteria were: (1) pain not controlled by opioids, (2) therapy with other PAMORAs, (3) intestinal obstruction with risk of intestinal perforation, and (4) severe renal insufficiency.

All patients provided written informed consent after being informed about the study and its aims.

No specific recommendations about the suspension or reduction of laxative therapy were given. The observation period lasted 4 weeks for each patient with 5 follow-up home visits by the palliative care physician: day 0 (baseline), day 7, day 14, day 21 and day 28.

Weekly data of the study were collected by the palliative care physician at the patient’s home in an electronic form appositely created by Vitaever^®^ SaaS (Nethical S.r.l., Bologna, Italy [27]), the Cloud technology tool used to record all the clinical information of patients assisted by ANT.

### 2.3. Endpoints

The primary endpoint was the assessment of constipation-related quality of life by the Patient Assessment of Constipation Quality of Life (PAC-QoL) questionnaire at baseline (day 0) and after 4 weeks (day 28) of naloxegol therapy [28]. PAC-QoL consists of 28 questions grouped into 4 subscales: worries and concerns, physical discomfort, psychosocial discomfort and satisfaction. The patients reported the severity of each symptom on a 0–4 point scale referred to the last 2 weeks. The score corresponding to each subscale was calculated by the sum of the scores of each item.

Secondary endpoints were: The objective constipation, assessed by the number of weekly spontaneous bowel movements (defined as a stool not induced by rescue medications) [29];The subjective perception of bowel function, assessed by the Bowel Function Index (BFI) [30], consisting of 3 numerical scales (from 0 to 100) to easily estimate OIC from the patient’s perspective: the difficulty of evacuation, the feeling of incomplete bowel evacuation and a personal judgment on constipation. The final score was calculated as the average of the scores obtained for each of the 3 scales. A BFI score greater than 30 indicated the presence of OIC [31]. The BFI was administered to patients every week;The intensity of the background pain, assessed by the Numerical Rating Scale (NRS) [32] every week.

### 2.4. Statistical Analysis

A descriptive analysis was executed for sociodemographic and clinical data of the patients. Qualitative variables were shown as frequency while quantitative variables were presented as mean ± St. Dev. According to the Shapiro-Wilk test, the variables related to the primary and secondary endpoints were not normally distributed, therefore, nonparametric statistical analysis was applied. The scores for PAC-QoL subscales were compared between day 0 and day 28 by the Wilcoxon signed-rank test. Monitoring of the number of weekly spontaneous bowel movements, BFI score and pain over time was performed by Friedman’s test for correlated sample adjusting the p obtained from pairwise comparisons by Bonferroni’s correction for multiple tests. A two-sided adjusted *p* < 0.05 was considered significant. Kaplan Meier survival curves were used to study the survival of patients from their entry into the study to a month after the conclusion of the recruitment period (January 2021). The comparison of age, KPS, duration and dose of opioid therapy, duration of laxative therapy, constipation-related quality of life, weekly spontaneous bowel movements, bowel function and pain at day 0 between patients completing the study and dropouts was analyzed by the Mann-Whitney U Test. The statistical analyses were executed by SPSS 25.0 for windows (SPSS Inc., Chicago, IL, USA).

### 2.5. Power Analysis

Considering the variation of the PAC-QoL subscale scores between day 0 and day 28 as primary outcomes, the effect size for the variation of each subscale was calculated. The effect sizes ranged from 0.42 (for the satisfaction) to 1.31 (for the physical discomfort). Given these effect sizes, a post hoc power analysis for a Wilcoxon signed-rank test with 78 patients was conducted, with the statistical power resulting from 95% to 99% in detecting the observed differences with a two-tailed alpha criterion of 0.05.

The power analysis was performed by G*power 3.1.9.4 (Franz Faul, Universität Kiel, Germany).

## 3. Results

### 3.1. Patients and Clinical Features

A total of 150 patients (77 men and 73 women, mean age 72.7 ± 11.1 years) were included in the study from 1 September 2018 to 31 December 2020 (Table 1). The majority of the patients were assisted by the ANT home palliative care program in Northern or Southern Italy (49.3% and 44.0%, respectively). 

At the entry, the mean Karnofsky Performance Score (KPS) was 48.9 ± 13.1 and the median of survival from the enrolment in the study was 83 days (95% C.I: 65–101 days). The most common primary sites of the tumor were gastrointestinal (27.3%), lung (22.0%), genitourinary (20.7%) and breast (10.0%). The majority of the patients had at least one metastasis (80.7%). The most frequent opioid therapies were fentanyl (52.6%) and oxycodone (24.7%). The mean daily morphine equivalent dose of the opioid was 42.1 ± 36.8 mg/die. The opioid therapy had been administrated from a mean of 82 ± 154 days. All the patients were in therapy with laxatives, mainly osmotic (44.0%) and combined (36.0%), for a mean duration of 56 ± 115 days.

Seventy-eight patients completed the 4 weeks of therapy with 25 mg/die of naloxegol. Among them, 44 patients (56.4%) continued the previously prescribed therapy with laxatives during the study: eight patients for the entire observation period, 15 for 3 weeks, 14 for 2 weeks and seven for 1 week. No changes in diet or physical activity were recorded during the follow-up.

### 3.2. Constipation-Related Quality of Life

An important improvement of all four dimensions of the constipation-related quality of life was observed in the patients completing the follow-up period (Figure 2). In particular, the scores obtained in the physical and psychological discomfort and worries/concerns subscales significantly decreased (*p* = 0.0009, *p* = 0.0019 and *p* = 0.0029, respectively), while the satisfaction level significantly increased (*p* = 0.0038). 

### 3.3. Bowel Function and Pain

Patients completing the 4 weeks of treatment had their first spontaneous bowel movement after 10 ± 16 h from the first dose of naloxegol. The number of weekly spontaneous bowel movements increased after a week from a baseline median value of 2 to 4 at day 7. In the next weeks, patients reported a median number of weekly spontaneous bowel movements of 5 (*p* < 0.001 for the comparisons between values at day 0 and at day 7, 14, 21 and 28) (Figure 3A).

The subjective perception of bowel function was considerably improved. The median BFI score decreased from 70 at day 0 to 30 at day 7 (*p* < 0.001), and it further decreased to 20 at day 28 (*p* < 0.001 for the comparisons between BFI scores at day 0 and at day 14, 21, and 28) (Figure 3B).

Finally, the intensity of the background pain was significantly reduced after a week (*p* < 0.001) and it remained lower during the following weeks (*p* < 0.001 for the comparisons between NRS scores at day 0 and at day 14, 21 and 28) (Figure 4).

### 3.4. Adverse Reactions

Patients completing the follow-up reported mainly gastrointestinal adverse reactions to naloxegol. The number of patients referring mild abdominal pain was high and increased during the observation period from 37 (day 7) to 51 (day 28), while a few patients reported moderate (*n* = 15) or severe abdominal pain (*n* = 6) at day 7, not worsening during the next weeks (Figure 5). Six patients experienced nausea/vomiting and 10 patients described other reactions (meteorism, lower back pain and headache) at day 7, but these effects tended to decrease during the follow-up period.

### 3.5. Dropout Patients

Seventy-two patients out of 150 suspended naloxegol therapy before day 28. Half of dropout patients interrupted the study during the first week (50.0%). This discontinuity was due to the following motivations: 21 patients died and 13 patients showed a severe clinical worsening due to their oncological disease during the observation period, 10 patients refused to continue for non-specified reasons, 9 patients were excluded by the physician due to non-adherence to the therapy, 5 patients interrupted the therapy due to the worsening of constipation (Table 2). Thirteen patients dropped out from the study for serious adverse reactions to naloxegol during the first week: 6 patients reported nausea, 3 moderate or severe abdominal pain, 2 diarrhea, 1 fecal incontinence and 1 opioid abstinence syndrome. The dropout patients showed a reduced median survival (38 days (95% C.I: 18–58)) in comparison to patients who completed the 4 weeks of follow up (125 days (95% C.I: 91–159)) (*p* < 0.001). No significant differences were observed between patients completing the study and dropouts regarding age, KPS, duration and dose of opioid therapy, duration of laxative therapy, constipation-related quality of life, weekly spontaneous bowel movements, bowel function and pain at the entry.

The analysis on patients who stopped therapy during the third or fourth week (*n* = 20) showed a significant improvement of weekly spontaneous bowel movements and BFI at day 7 and day 14 (*p* < 0.001) (Figure 6A,B). No significant difference was observed about the intensity of the background pain (Figure 6C).

## 4. Discussion

The home palliative care program managed by the ANT Foundation takes into great consideration the quality of life of the patients with advanced cancer, offering the opportunity to live the last period of life in one’s home environment with comprehensive multidisciplinary assistance to control pain and other symptoms, as well as to support the patient with physical and psychological difficulties [33,34].

Constipation exerts a debilitating effect on patients receiving opioids for cancer pain and strongly impairs their quality of life. A correct evaluation and treatment of OIC is essential, and when the laxative therapy is ineffective, a target therapy with PAMORAs should be considered [20]. Naloxegol has been indicated as one of the most effective PAMORAs to improve/resolve OIC without reducing the doses of opioid therapy [35,36,37]. The 25 mg/day dose of naloxegol has been reported as the most efficacious, safe and well tolerated in patients with OIC [37,38,39]. Some trials showed that naloxegol improves straining, stool consistency and spontaneous bowel movements in non-cancer patients with OIC [35,36,37] but, until now, only the KYONAL study evaluated its effect in cancer patients followed for 3 months [23] and for 1 year [24]. In addition, another PAMORA, the naldemedine, is orally available and seems to be efficient in increasing the number of bowel movements [40], but it was not yet available in the EU at the start of the study.

Our study evaluated the impact of naloxegol in improving or resolving OIC [21] in advanced cancer patients no longer susceptible to specific cancer therapy. The possibility to improve the quality of life in these patients, even if they had a poor life expectancy, was the primary endpoint of this research. The results obtained from the 78 patients completing the 4 weeks of follow up confirmed the efficacy of naloxegol in contrasting the opioid’s inhibitory action on gastrointestinal motility, with an improvement in objective and subjective symptoms of OIC. A noteworthy improvement of the quality of life was observed. Patients reported a significant reduction of physical and psychological discomfort, as well as of worries and concerns associated with constipation. At the same time, the level of satisfaction results remarkably increased. The heterogeneity of the sample population analyzed may lead to the high variance of the outcome measures. Although there are not enough data to demonstrate a direct correlation with naloxegol assumption, patients referred a significant reduction of the background pain, thus suggesting a likely relationship. These results are similar to those found by the KIONAL study [23,24] although, in our study, naloxegol exerted more severe, frequent and persistent side effects. The majority of the patients reported abdominal pain, mainly of mild intensity, for the entire duration of the treatment period. This finding could be related to the worst clinical conditions and a lower KPS due to a more advanced stage of cancer in respect to the KIONAL patients. In fact, in our study, patients completing the study showed a median survival of 125 days. The reduction of the background pain resulting in psycho-physical benefits suggests that naloxegol therapy, even in the short 4-week period, may be effective in improving the quality of life in these patients.

The number of dropout patients was considerably high. The median life expectancy of these patients was 38 days resulting significantly lower compared to the survival of patients completing the study. Almost half of the cases dropped out due to death or clinical worsening caused by cancer complications. Adverse reactions occurred mainly during the first week, confirming the findings from Dols et al [23].

The nine patients excluded from the present study for non-adherence to the therapy used naloxegol as a “rescue” laxative rather than a target therapy to be taken daily according to the indicated therapeutic scheme. The patient’s self-interruption of naloxegol treatment once evacuation has occurred is frequently reported by literature [21,41]. 

However, patients who followed naloxegol therapy for at least 2 weeks showed a significant improvement of bowel function.

The positive impact of the improvement/resolution of the OIC on the quality of life, even in patients with advanced cancer, emphasizes the importance of the early and accurate identification of the constipation for an appropriate and immediate choice of the most suitable treatment. The high dropout rate due to worsening clinical conditions and adverse reactions suggests that the choice of the drug to treat the OIC should consider the predictive survival prognosis as a therapeutic line selection criteria.

Despite the many strengths, the present paper has also some limitations. This is an observational study conducted under conditions of standard clinical practice without a control arm. In our research setting, involving advanced cancer patients in home palliative care, large randomized clinical trials are challenging and not ethically possible [42]. The results of this study on patients assisted at home with a 2–3 month life expectancy may be barely generalizable to all cancer patients. Finally, the evaluation of the constipation focused on the bowel function and did not assess the quality and/or type of the stools.

## 5. Conclusions

Naloxegol was effective in resolving OIC, reducing background pain and improving the quality of life in patients with advanced cancer in home palliative care. Even in patients with a very short life expectancy, naloxegol therapy improved bowel function within 2 weeks.

## Figures and Tables

**Figure 1 cancers-13-05736-f001:**
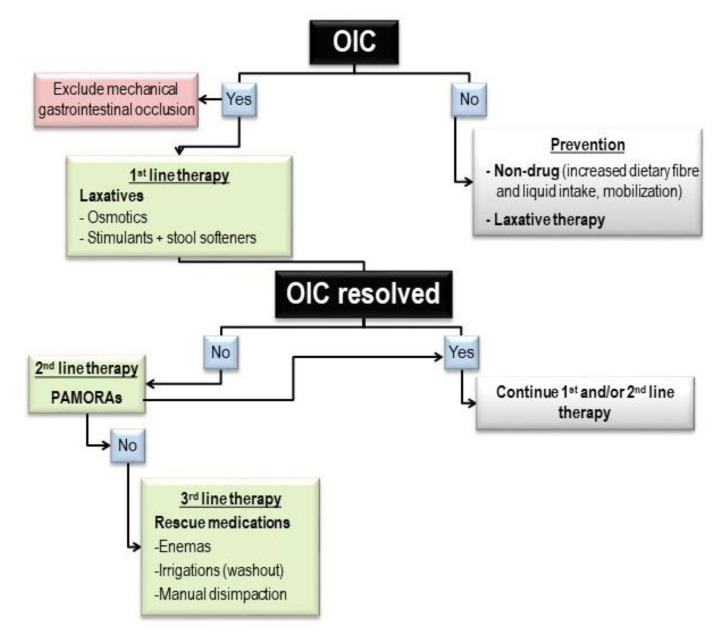
Treatment of constipation in cancer patient on opioid therapy.

**Figure 2 cancers-13-05736-f002:**
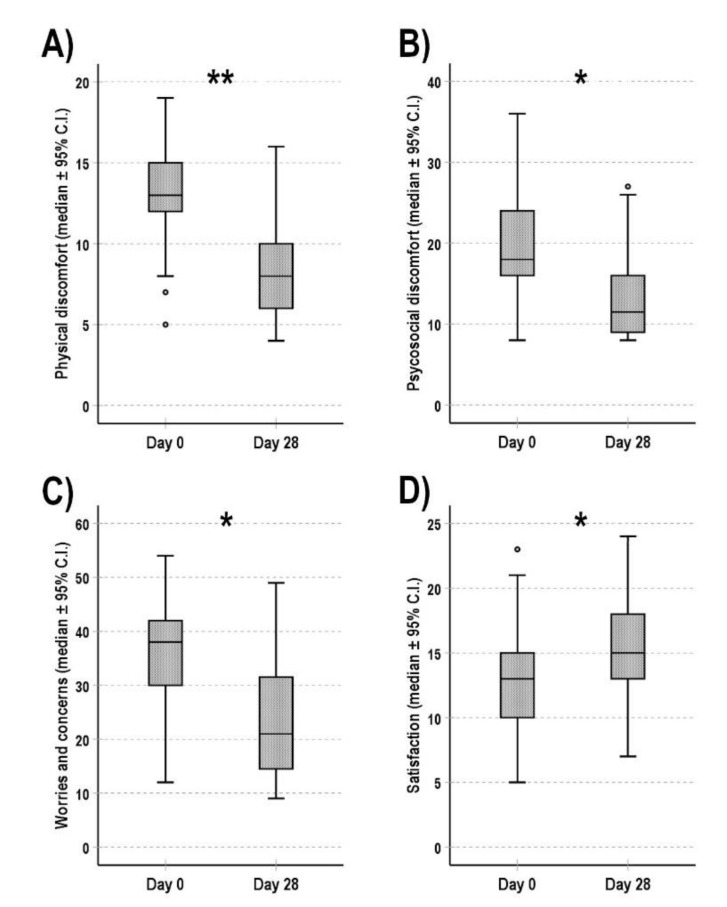
Constipation-related quality of life at the baseline (day 0) and after 4 weeks of follow up (day 28). The scores obtained in the four dimensions of PAC-QoL questionnaire ((**A**) physical discomfort, (**B**) psychological discomfort, (**C**) worries/concerns and (**D**) satisfaction) are shown as median (95% C.I.). * *p* < 0.010; ** *p* < 0.001.

**Figure 3 cancers-13-05736-f003:**
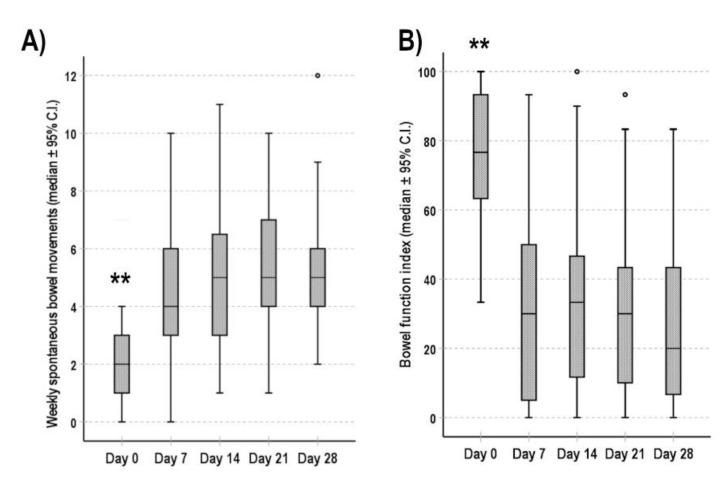
Evolution of bowel function during the 4 weeks of follow up. (**A**) Number of weekly spontaneous bowel movements, (**B**) BFI score. Data are shown as median (95% C.I.). ** *p* < 0.001 for the comparisons between values at day 0 and values at day 7, 14, 21 and 28.

**Figure 4 cancers-13-05736-f004:**
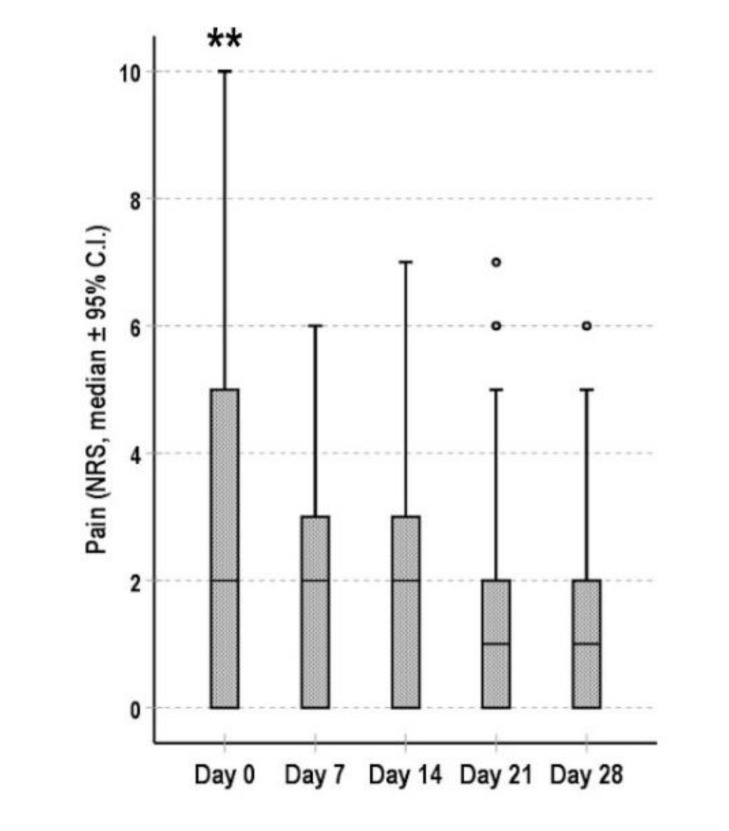
Evolution of pain during the 4 weeks of follow-up. Data are shown as the median (95% C.I.) of the NRS score. ** *p* < 0.001 for the comparisons between values at day 0 and values at day 7, 14, 21 and 28.

**Figure 5 cancers-13-05736-f005:**
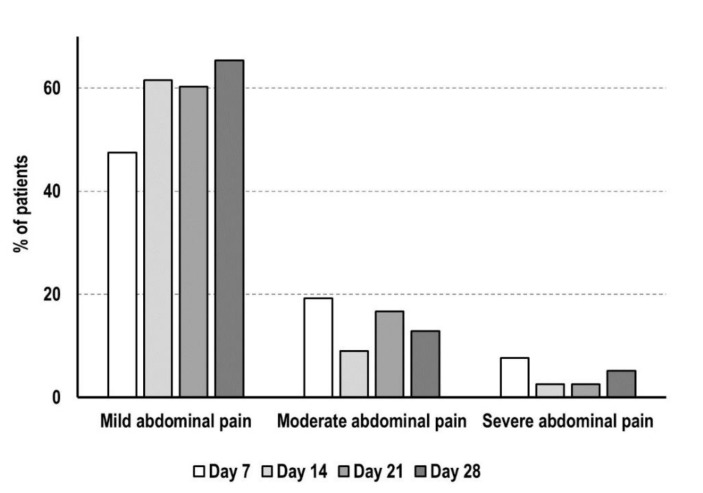
Abdominal pain in patients completing the 4 weeks of follow-up. Data are shown as percentage of patients reporting the adverse reaction to naloxegol for each week.

**Figure 6 cancers-13-05736-f006:**
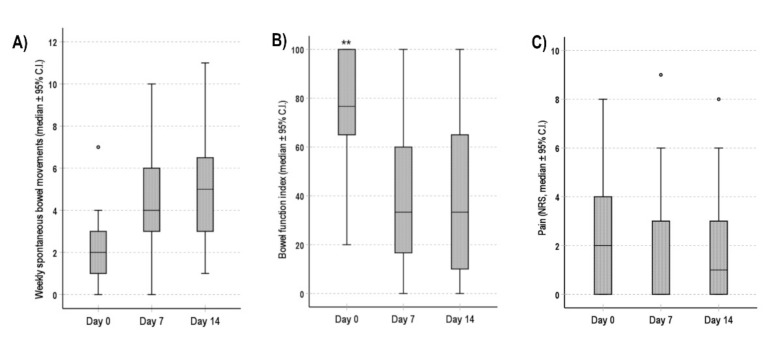
Bowel function and pain in dropout patients (*n* = 20) during the first 2 weeks of follow-up. (**A**) Number of weekly spontaneous bowel movements, (**B**) BFI score, (**C**) Pain (NRS score). Data are shown as median (95% C.I.). ** *p* < 0.001 for the comparisons between values at day 0 and values at day 7 and 14.

**Table 1 cancers-13-05736-t001:** Sociodemographic and clinical data of the patients at the entry.

Patients, *n* (%)	150
Men	77 (51.3%)
Women	73 (48.7%)
Age (mean ± St. Dev.)	72.7 ± 11.1
Geographical area, *n* (%)	
Northern Italy	74 (49.3%)
Central Italy	10 (6.7%)
Southern Italy	66 (44.0%)
KPS (mean ± St. Dev)	48.9 ± 13.1
Survival (days), (median; 95% C.I.)	83 (65–101)
Tumor primary site, *n* (%)	
Gastrointestinal	41 (27.3%)
Lung	33 (22.0%)
Genitourinary	31 (20.7%)
Breast	15 (10.0%)
Head/neck	9 (6.0%)
Other ^a^	21 (14.0%)
Metastasis, *n* (%)	121 (80.7%)
Opioid therapy, *n* (%)	
Fentanyl	79 (52.6%)
Oxycodone	37 (24.7%)
Buprenorphine	10 (6.7%)
Morphine	12 (8.0%)
Other ^b^	12 (8.0%)
Morphine equivalent dose, mean ± St. Dev. (mg/die)	42.1 ± 36.8
Duration of opioid therapy, mean ± St. Dev. (days)	82 ± 154
Laxative therapy, *n* (%)	
Osmotic	66 (44.0%)
Combined	54 (36.0%)
Emollient	17 (11.3%)
Stimulant	13 (8.7%)
Duration of laxative therapy, mean ± St. Dev. (days)	56 ± 115

^a^ Central nervous system, hematological, pancreas, skin, sarcoma; ^b^ Tapentadol, tramadol, codeine *plus* paracetamol.

**Table 2 cancers-13-05736-t002:** Motivation for dropping out in the study population.

Motivation for Dropping Out	Total, *n* (%)	Within Day 7	Within Day 14	Within Day 21	Within Day 28
Death	21 (29.2%)	9	6	4	2
Severe clinical worsening	13 (18.0%)	3	3	7	/
Serious adverse reaction	14 (19.4%)	13	/	/	1
Refusal for other reasons	10 (13.9%)	7	2	1	/
Non-adherence to therapy	9 (12.5%)	4	3	/	2
Worsening of constipation	5 (7.0%)	/	2	2	1

## Data Availability

The data presented in this study are available on request from the corresponding author.

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
