# Peer review of "Can Naloxegol Therapy Improve Quality of Life in Patients with Advanced Cancer?"

_cancers, 2021, doi:10.3390/cancers13225736_

Round 1
Reviewer 1 Report
Dear Authors
The authors evaluated the efficacy of naloxegol in resolving opioid-induced constipation (OIC) and in improving the quality of life of advanced cancer patients with a short life expectancy in a home palliative setting.
The authors concluded that naloxegol improved the quality of life, resolved OIC, and reducing overall pain.
This article is well-written and informative for the readers.
I have one comment.
The authors choiced naloxegol as peripheral acting μ-opioid receptor antagonists (PAROMA) for resolving OIC.
Other kind of PAROMA, such as naldemedine, should be discussed in discussion.
Best regards
Author Response
We greatly appreciate your positive judgement.
According your suggestion, we added a comment in the introduction about the choice of naloxegol (lines 58-61) and a comment about naldemedine in the discussion (lines 260-263).
Thank you very much for your careful revision!
Reviewer 2 Report
This is an observational study of the effect of naloxegol on reducing opioid-induced constipation (OIC) and improving quality of life in cancer patients with OIC. The following comments and suggestions are offered in order to improve an already well-written manuscript.
Title: Consider changing the title to "Can Naloxegol Therapy Improve Quality of Life in Patients with Advanced Cancer?"
Abstract:
Key words: PAMORA isn't a MeSH term. Consider 'antagonist, opioid receptor.'
Introduction: At line 43, consider using gender neutral language. At line 58, the study also addressed whether constipation was reduced in addition to QoL.
Methods: There was no definition of "advanced cancer" or how "maintaining their mental capacity" was demonstrated or what was a "spontaneous bowel movement." There is no description of how patients were recruited (seems like a convenience sample) nor the quality and/or type of the bowel movement (there are 7 types according to the Bristol Stool Chart). There is no sample size calculation. The study was not intention-to-treat, so we do not know how many patients need to be treated in order for a significant result to be achieved. Dropouts were analyzed separately, instead of being included, which was noted as a study weakness by the authors. Why were dropouts included in part of the analysis, and not all of it? This introduces bias.
Results: There is no discussion of the meaning of KPS score. There was huge variance in many of the outcome measures, which leads me to believe that this was a very heterogenous sample.
Conclusion: The statement: "In patients with a very short life expectancy an evaluation of risk / benefit should be carefully considered for the choice of the most effective therapeutic line" while perhaps commonsense, does not follow from the results, and needs to be omitted. Focus on what the study actually found or speculate on what you would do in the future to improve on the quality and generalizability of the findings.
References: not in mdpi style.
Provided these issues can be resolved, I would recommend re-consideration for publication. Thank you for the opportunity to review your manuscript.
Author Response
We greatly appreciate your comments that, in our opinion, have been very helpful to improve our work. We tried to do our best to revise the manuscript according to your recommendations.
Please consider the following revisions we did according your suggestions:
Title: Consider changing the title to "Can Naloxegol Therapy Improve Quality of Life in Patients with Advanced Cancer?"
Key words: PAMORA isn't a MeSH term. Consider 'antagonist, opioid receptor.'
Done! We changed the title and keyword according your proposal.
Introduction: At line 43, consider using gender neutral language. At line 58, the study also addressed whether constipation was reduced in addition to QoL.
Done! We corrected “using gender neutral language” at line 45 and added the reduction of constipation to the aim of the study (line 63).
Methods: There was no definition of "advanced cancer" or how "maintaining their mental capacity" was demonstrated or what was a "spontaneous bowel movement."
We added the definition of “advanced cancer” (lines 78-79), the assessment of mental capacity (lines 88-89) and the definition of “spontaneous bowel movement” (lines 115, reference 29).
There is no description of how patients were recruited (seems like a convenience sample) nor the quality and/or type of the bowel movement (there are 7 types according to the Bristol Stool Chart).
We tried to better explain the recruitment process (lines 76-104).
Unfortunately, we did not assess the type of stool by the Bristol Stool Chart but we focused on the bowel function. We kindly ask the reviewer to consider the difficulties in the data collection due to the home setting and the fragile condition of the patients. We added this point as a limitation of the study (lines 309-310).
There is no sample size calculation. The study was not intention-to-treat, so we do not know how many patients need to be treated in order for a significant result to be achieved.
Thank you for this observation. We performed a post hoc power analysis for a Wilcoxon signed-rank test with the 78 patients who completed the observation period. The statistical power resulted from 95% to 99% in detecting the observed differences in the primary outcomes (variation of the PAC-QoL subscales) (lines 142-149).
Dropouts were analysed separately, instead of being included, which was noted as a study weakness by the authors. Why were dropouts included in part of the analysis, and not all of it? This introduces bias.
We decided to analyse the dropouts separately because their number was considerably high (almost the half of the patients enrolled). In the first version of the paper, we tried to characterize in depth this group of patients showing that, although they were similar to the patients completing the study at the entry (age, KPS, opioid and laxative therapy), they showed a reduced survival. According your observation, in the revised version of the manuscript, we added some details about these patients in order to describe them even better: the comparison at day 0 of constipation-related quality of life, weekly spontaneous bowel movements, bowel function and pain between dropouts and patients completing the study was analysed by Mann–Whitney U test (lines 136-139) and no significant differences were observed (lines 229-232). Moreover, we added the graph of the intensity of the background pain (figure 6, panel C) in dropouts. For dropouts, we showed the results of weekly spontaneous bowel movements, BFI and pain only at day 0, day 7 and day 14 omitting the day 21 for the low number of patients remaining. Regarding the quality of life, the PAC-QoL questionnaire was administered at the entry (day 0) and at the end (day 28) of the observation period, so we skipped the graph for the dropouts because we did not have the data at day 28.
Results: There is no discussion of the meaning of KPS score.
KPS has not been considered as inclusion /exclusion criteria nor as an endpoint. We collected and reported this fundamental clinical data to characterize the functional status of the enrolled patients. We used KPS at the entry to compare functional status between patients completing the study and dropouts (line 230). In addition, we discussed KPS to explain the different results obtained respect to the KYONAL study (line 281).
There was huge variance in many of the outcome measures, which leads me to believe that this was a very heterogeneous sample.
Thank you, we agree to this point and we added a comment in the discussion (lines 273-274).
Conclusion: The statement: "In patients with a very short life expectancy an evaluation of risk / benefit should be carefully considered for the choice of the most effective therapeutic line" while perhaps commonsense, does not follow from the results, and needs to be omitted. Focus on what the study actually found or speculate on what you would do in the future to improve on the quality and generalizability of the findings.
Thank you for this observation. We modified the discussion (lines 301-303) and conclusions according your suggestion (lines 315-316).
References: not in mdpi style.
We corrected the style of the references.
Provided these issues can be resolved, I would recommend re-consideration for publication. Thank you for the opportunity to review your manuscript.
Thank you very much for your careful revision!